# Antifungal Potential of *Azotobacter salinestris* Strain *Azt* 31 against Phytopathogenic *Fusarium* spp. Associated with Cereals

**DOI:** 10.3390/jof8050473

**Published:** 2022-04-30

**Authors:** Hanumanthu Nagaraja, Gurikar Chennappa, Nagaraj Deepa, Manjunath Krishnappa Naik, Kadaiah Ajithkumar, Yatgal Sharanappa Amaresh, Premila N. Achar, M. Y. Sreenivasa

**Affiliations:** 1Department of Studies in Microbiology, University of Mysore, Manasagangotri, Mysore 570006, India; nagarajh82@gmail.com (H.N.); chinnagurikar@gmail.com (G.C.); deepanagraju@gmail.com (N.D.); 2Department of Food Technology, Ramaiah University of Applied Sciences, Bangalore 560054, India; 3Department of Plant Pathology, University of Agricultural Sciences, Raichur 584104, India; mknaikuasr@gmail.com (M.K.N.); ajithk.path@gmail.com (K.A.); amaresh.ys@gmail.com (Y.S.A.); 4Department of Molecular and Cellular Biology, Kennesaw State University, Kennesaw, GA 30144, USA

**Keywords:** *Azotobacter*, biocontrol, maize, sorghum, wheat, *Fusarium*, fungal infection, seed treatments

## Abstract

Antifungal efficacy of *Azotobacter salinestris* against trichothecene-producing *Fusarium* spp. was investigated in maize, sorghum, and wheat. The three cereals were subjected to four treatments as control (T1), *Fusarium* alone (T2), combination of *Fusarium* and *A. salinestris* treatment (T3), and only *A. salinestris* (T4). All the treatments were evaluated for total mass of seedlings, root and shoot length, seed germination, and vigor index (VI), and extent of rhizoplane colonization by *A. salinestris* was investigated. Further, greenhouse studies were conducted to learn the efficacy of *A. salinestris* in vivo conditions. Antifungal efficacy was tested by the dual-culture method which resulted in significant reduction in *Fusarium* growth. Infection by *Fusarium* was reduced up to 50% in treated cereals such as maize, sorghum, and wheat, and there was also significant increase in seedling mass in the three hosts. Maize showed the highest VI (1859.715), followed by sorghum (1470.84), and wheat (2804.123) with *A. salinestris* treatment. In addition, seed germination was enhanced to 76% in maize, 69% in sorghum, and 68% in wheat, respectively. Efficacy of rhizoplane colonization showed successful isolation of *A. salinestris* with high CFU rate, and furthermore, significant colonization inhibition by *Fusarium* spp. was observed. In the greenhouse conditions, on the 45th day of the experimental set-up, the highest shoot length/root length recorded in maize was 155.70/70.0 cm, in sorghum 165.90/48.0 cm, and in wheat 77.85/56.0 cm, and the maximum root mass recorded was 17.53 g in maize, 4.52 g in sorghum, and 1.90 g in wheat. Our present study showed that seed treatment by *A. salinestris,* may be used as an alternate biocontrol method against *Fusarium* infection in maize, sorghum, and wheat.

## 1. Introduction

Cereal crops are an essential nutritional source for the worldwide population, and they are of great economic importance both as food and feed [1]. Grain industries generate many cereal-based products. Higher incidence of infection in cereals occurs either in the field or during post-processing of the food grains, which has become a threat for the food market’s ability to produce quality products. Toxigenic fungal pathogens including *Fusarium*, *Aspergillus*, and *Alternaria* spp. are responsible for many diseases and economic losses [2].

Among the different fungal pathogens, *Fusarium* spp. are widespread pathogens affecting grains (maize, sorghum, wheat, barley, oats, rye, etc.) and this results in reduction of crop yield to the extent of 10–40% worldwide [3]. *Fusarium* spp. are either seed- or soil-borne, causing diseases such as ear rot in maize, stalk rot in wheat, and head blight (scab) in small-grain cereals. In addition to their direct impact on cereals, *Fusarium* spp. can also produce mycotoxins such as deoxynivalenol (DON), fumonisins, moniliformin, fusarins, and zearalenone in the infected ears and kernels [1,4]. *Fusarium* toxins are known to possess carcinogenic and toxigenic properties in farm livestock and humans [5]. There is evidence proving that mycotoxin problems have resulted in destruction of crops, leading to significant imbalance in the food chain, ultimately affecting the economy of countries worldwide [6]. Among the *Fusarium* toxins, trichothecene is the major mycotoxin, commonly found in cereals and cereal-based foods. Colonization of cereals by trichothecene-producing *Fusarium* spp. on grains and food products causes discolorations, decreased germination and vigor, heating, mustiness, dry matter loss of the grain, and deterioration in nutritional quality, ultimately resulting in economic loses [7]. *Fusarium* spp. colonization in cereals, feed, and related products is also associated with potential human and animal health risks and leads to food-borne intoxication, especially among farm animals [8]. Early detection of phytopathogens growth and their control in cereals during both pre- and post-harvest is of great importance [9,10].

Different fungicides and chemical agents are available in the market to control infection of cereals by *Fusarium* spp. However, long-term use of synthetic fungicides has many disadvantages: gradual aggravation of soil fertility, development of pathogen resistance to fungicides, inability to reach the roots of mature plants, rapid degradation in soil, and the need for repeated applications are associated with the use of fungicides and chemicals as antifungal agents [11]. Therefore, there is a need for the development of alternative and environmentally safe methods of control against toxigenic fungal pathogens and to increase yields while producing quality cereals. In recent years, biological control of *Fusarium* spp. using antagonistic bacteria is gaining importance, since this biological control agent is sustainable, safe, and environmentally friendly [12,13,14].

Plant-growth-promoting rhizobacteria (PGPR) are a heterogeneous group of bacteria that enhance plant growth by direct or indirect mechanisms. They are present in the rhizosphere, at root surfaces, and are also associated with soil-borne pathogen restrictions by induction of systemic resistance, phytohormone production, and quorum sensing [15]. These bacteria also promote beneficial plant–microbe symbioses and interfere with toxins production by the pathogens [15]. A large variety of bacteria including species belonging to *Pseudomonas*, *Azospirillum*, *Azotobacter*, *Klebsiella*, *Alcaligens*, *Arthobacter*, *Bacillus*, *Enterobacter, Lysobacter*, and *Serratia* are reported to improve plant growth [16,17,18]. Understanding the diversity of PGPR in the rhizosphere as well as their colonization ability and mechanism of action makes them useful in management practices [19]. Accordingly, post-harvest *Penicillium* spp. and *Rhizopus stolonifera,* in tomato disease management by PGPR *Bacillus subtilis*, were reported by Punja et al. [20].

Among PGPR, the genus *Azotobacter* is currently being applied in agricultural crops for plant growth enhancement, seed germination, and control of soil-borne pathogens. *Azotobacter* spp. can tolerate extreme environmental conditions and can survive by producing cysts. *Azotobacter salinestris*, *Azotobacter vinelandii*, and *Azotobacter chroococum* are a few well-known species isolated and identified from different types of soil. These organisms have also been reported to degrade pesticides, fix atmospheric nitrogen (N_2_), and produce plant growth hormones such as indole acetic acid and gibberellic acid, and are known for their antagonistic activity against phytopathogenic fungi, which helps in disease management of various crops [13,14]. 

The present study was carried out with the following objectives: (a) to investigate *A. salinestris* efficacy in controlling *Fusarium acuminatum*, *Fusarium avenaceum*, *Fusarium crookwellense*, *Fusarium culmorum*, *Fusarium equiseti*, *Fusarium graminearum*, *Fusarium nivale*, *Fusarium poae*, *Fusarium sambucinum*, and *Fusarium sporotrichioides* which are known for trichothecene production in maize, sorghum, and wheat; (b) to evaluate the efficacy of *A. salinestris* in improving total mass of seedlings, root and shoot length, and seed germination (VI) in maize, sorghum, and wheat grains; and (c) to evaluate the extent of rhizoplane colonization by *A. salinestris* in maize, sorghum, and wheat treated with trichothecene-producing *Fusarium* spp.

## 2. Materials and Methods

### 2.1. Strain and Culture Conditions

*Fusarium* spp. such as *F. sporotrichioides*, *F. crookwellense*, *F. graminearum*, *F. poae*, *F. sambucinum*, *F. culmorum*, *F. acuminatum*, *F. avenaceum*, *F. nivale*, and *F. equiseti* as shown (Table 1), were isolated from cereals such as maize (*Zea mays*), sorghum (*Sorghum bicolor*), and wheat (*Triticum aestivum*) collected from different geographical areas in Karnataka state, India. All the *Fusarium* isolates were identified up to the species level by microscopic and molecular methods. The ability of *Fusarium* isolates to produce trichothecenes was confirmed by thin-layer chromatography (TLC) [21] and liquid chromatography mass spectrometry (LCMS). 

*A. salinestris* strain AZT 31 was isolated from soil samples from a paddy field. The strain was identified to the species level using the above methods. The identification was further confirmed by DNA barcoding using part of the 16S ribosomal DNA. The sequence reads were submitted to Gen Bank, NCBI, with accession No JX262176 [22].

The isolated fungal and bacterial species were maintained in glycerol stocks at 4 °C. Fungal isolates were subcultured on potato dextrose agar (PDA) and *A. salinestris* on Waksman selective media (10.0 g/L Mannitol, 5.0 g/L CaCO_3_, 0.5 g/L K_2_HPO_4_, 0.2 g/L MgSO_4_, 0.2 g/L NaCl, traces of MnSO_4_, FeCL_3_ 20 g/L Agar).

### 2.2. Inoculum Preparation

#### 2.2.1. Bacterial Cell Suspension

Waksman broth (50 mL) was inoculated with two loops full of 3-day-old *A. salinestris* culture and incubated for 48 h at 34 °C. After incubation, bacterial cells were collected by centrifugation (7155 g, 15 min, 4 °C) and were resuspended in 1X phosphate saline buffer (137 mM NaCl, 2.7 mM KCl, 10 mM Na_2_HPO_4_, 1.8 mM KH_2_PO_4_) at pH 7.2. Bacterial cell suspension was adjusted to 10^9^ CFU/mL by following standard serial dilution and absorbance studies.

#### 2.2.2. Fungal Spore Preparation

In total, 100 mL of potato dextrose broth (PDB) was prepared in a 250 mL Erlenmeyer flask for sporulation of all tested fungal strains. A fungal disk (5 mm) from pure culture of each *Fusarium* sp. was inoculated into the PDB flask using a sterile cork borer [3]. The flasks containing the PDB cultures were incubated at 25 ± 2 °C for 4 days on a shaker incubator (100 rpm) for 12 h, with alternate light and dark conditions. Fungal spores were harvested, and spore concentration was standardized to 10^7^ spores/mL in 5% NaCl using a hemocytometer [23].

### 2.3. Seed Treatment

Maize, sorghum, and wheat grains were treated with sodium hypochlorite as the method described by Nagaraja et al. [21]. Fifty grams of each seed sample was surface-sterilized with 4% sodium hypochlorite followed by washing with sterile water to remove surface contaminants. Each seed sample was subjected to four different treatments as mentioned (Table 2). Each seed sample was soaked in 50 mL of sterile water in 100 mL Erlenmeyer flask, which acted as control (T1). Treatments T2 and T4 consisted of soaking each seed sample with 50 mL of fungal spore suspension and *A. salinestris* cell suspension, respectively. Treatment T3 was conducted by soaking each seed sample in a mixture of 25 mL each of bacterial cell and fungal spore suspensions, in 100 mL Erlenmeyer flasks. All the treated samples were incubated for 2 h at 28 °C in a shaker incubator, at 100 rpm, for uniform coating of the tested organisms (Table 1 and Table 2). The experiment was conducted in duplicates.

### 2.4. In Vitro Studies of A. salinestris against Fusarium *spp*.

#### 2.4.1. Antagonistic Activity of *A. salinestris* Investigated by Dual-Culture Method

Antagonistic efficiency of *A. salinestris* against mycelial growth of phytopathogenic *Fusarium* spp. as indicated (Table 1) was studied by measuring the zone of inhibition on modified Waksman media and PDB at 1:1, *v*/*v*, following the modified protocol of Daniel et al. [24]. *A. salinestris* culture at cell concentration of 10^9^ CFU/mL was streaked horizontally at the center of the Petri plates and incubated at 24 ± 2 °C, for 42 h. After the growth of bacterial colonies, a 5 mm agar disk from a fresh culture of *Fusarium* spp. (Table 1) was placed perpendicularly on either side of the bacterial margin. Plates without the antagonist bacteria and inoculated with the fungal agar disc served as control. All treated and untreated plates were incubated at 24 ± 2 °C for 3 to 5 days. The zone of inhibition was measured using a ruler from the edge of the bacterial margin to the edge of each respective fungal colony. The experiment was carried out in triplicates. 

#### 2.4.2. Effect of *Azotobacter* against *Fusarium* spp. and Infection Incidence

This assay was conducted to study the influence of *A. salinestris* in controlling the severity of *Fusarium* spp. in these cereals. Infection incidence (%) in maize, sorghum, and wheat discovered by using the standard blotter method is indicated (Table 1). One hundred seeds from each sample were surface-sterilized and subjected to T1, T2, T3, and T4 treatments (Table 2). Seeds were placed in sterilized Petri plates and incubated at 23 ± 2 °C for 5 days. Treated seeds were visualized for fungal growth using a stereo-binocular microscope and a compound microscope (Labomed). Further infection incidence was confirmed by visualizing their morphological characteristics such as micro conidia, macro conidia, and chlamydospores, respectively, with reference to Leslie and Summerell. The infection was calculated using the standard formula [21]. Experiment was carried out in triplicates. 

#### 2.4.3. Effect of *A. salinestris* against Vigor Index of Maize, Sorghum, and Wheat 

The VI parameter of maize, sorghum, and wheat grains was studied using the paper towel method. Fifty seeds from each cereal grain were subjected to T1, T2, T3, and T4 treatments as shown (Table 2), and placed separately on a moist germination sheet, overlaid with another layer of similar sheet (31 cm W × 45 cm L grade 1 paper towel) and rolled tightly. The experimental set-up was carried out in duplicates. Tightly rolled sheets were incubated at an inclined position in a growth chamber for 7 days at 28 °C. Moisture was provided to avoid drying of seedlings. The experiment was carried out in triplicates. The VI growth such as root and shoot length parameters, percentage of germination, root length, shoot length (cm), and total seedlings mass (grams) were recorded and the VI was calculated using the formula [17]: VI = (mean root length + mean shoot length) × % germination.

### 2.5. Effect of A. salinestris against Root Colonization of Cereal by Fusarium *spp*.

Twenty seeds of maize, sorghum, and wheat from T1, T2, T3, and T4 treatments (Table 2) were each sown in a germination tray, each containing 50 cups (plastic trays of 5 cm × 5 cm size) filled with field soil (mixture of red soil, sand, and brown soil) and coco peat (procured from the nursery). Each tray was incubated in a growth chamber at 28 °C for 20 days with 12 h of light and 12 h of dark, with the addition of frequent sterilized distilled water. Following incubation, 20-day-old germinated seedlings were uprooted, and one gram of roots representing the entire rhizospheric region along with its soil was used for analysis. The roots were washed in 10 mL of sterile PBS for 15 min, and eluant was collected. A total of 1 mL of PBS from the collecting tube was serially diluted up to 10^−6^ dilution. In total, 100 µL of the 10^−6^ dilution sample was inoculated onto Waksman agar and incubated at 34 °C for 24–48 h for expression and isolation of *Azotobacter.* In total, 100 µL of 10^3^ dilutions was inoculated in 2.5 ppm malachite green agar (MGA) 2.5 ppm and 100 µL of 10^3^ dilutions was inoculated in potato dextrose chloramphenicol agar (PDCA), which were incubated as mentioned above for isolation of *Fusarium* spp. [25]. The microscopic identifications were performed by observing its micro and macro conidia and chlamydospores for respective fungal pathogens according to Leslie and Summerell [26]. The total colony counts of bacteria were performed by the plate count method using a cubic colony counter and were expressed as Log_10_ CFU g^−1^ for each cereal root sample. The experiment was carried out in triplicates.

### 2.6. In Planta Antagonistic Activity of A. salinestris against Fusarium *spp*.

Based on our in-vivo experimental data, five prominent cereal contaminants of *Fusarium* spp. such as *F. sporotrichioides, F. crookwellense, F. graminearum, F. poae*, and *F. culmorum* were selected for the greenhouse studies. Ten seeds from wheat, maize, and sorghum were randomly selected and treated with *A. salinestris* by seed coating, followed by planting at 4 cm deep in sprouting nursery poly bags (10 cm W × 30 cm L) filled with 3/4th of field soil. Each bag was topped with 25 g of coco peat, a natural fiber made of coconut husk, consistent and easy to handle. The coco peat also serves as a very good seed sowing medium which retains moisture up to eight times its volume and acts as a perfect soil conditioner. Additional moisture content for the germination process was maintained by regular watering. The shoot length of the three treated cereal samples in each treatment was measured at growth intervals of 15, 30, and 45 days after sowing. Untreated samples served as control (Table 2). After forty-five days of seed sowing, treated and untreated plants were uprooted, and their root length and total root mass were measured. The experiment was repeated in duplicates.

### 2.7. Statistical Analysis

All growth parameters of maize, sorghum, and wheat in different treatments such as in vitro and in vivo studies were analyzed by one-way analysis of variance (ANOVA). The software used was SYATAT-Sigma Stat for Windows version 3.1 (SPSS, 2.0). Significant difference was determined at *p* ≤ 0.05. Mean and standard errors represented are the values of triplicates. Graphs were constructed with the GraphPad Prism 5 program. 

## 3. Results 

### 3.1. In Vitro Studies of A. salinestris against Fusarium *spp*.

#### 3.1.1. Antagonistic Activity of *A. salinestris* Investigated by Dual-Culture Method

In the present study, the anti-fungal efficacy of *A. salinestris* in controlling trichothecene-producing *Fusarium* species was investigated using the dual-culture technique, as a preliminary study. Our assay showed that among the ten *Fusarium* spp. tested (Table 1), the growth of five *Fusarium* species, namely *F. sporotrichioides, F. crookwellense, F. graminearum, F. poae,* and *F. culmorum* was inhibited by the presence of *A. salinestris* (Figure 1) based on the fungal growth and zone of inhibition. 

#### 3.1.2. Effect of *Azotobacter* against *Fusarium* spp. and Infection Incidence in Cereals

The ability of *A. salinestris* to reduce the growth of *Fusarium* spp. and colonization in maize, sorghum, and wheat was tested by the infection incidence (%) assay. Our results showed that *A. salinestris* significantly decreased the *Fusarium* spp. infection. Infection incidence in grains (T2) was recorded up to 90–96% in maize, 91–98% in sorghum, and 84–90% in wheat. Treatment with *A. salinestris* (T3) significantly reduced the infection incidence up to 35.5% in all three cereals. In maize, infection incidence was more effective by *F. crookwellense* and *F. culmorum*, with an infection incidence up to 20 and 23%, respectively. In sorghum, infection incidence by *F. sambucinum* was reduced by 26% when treated with *A. salinestris.* In wheat seedlings, when treated with *A. salinestris*, infection incidence by *F. sambucinum* was reduced by 29% (Figure 2). 

#### 3.1.3. Effect of *A. salinestris* against Vigor Index of Maize, Sorghum, and Wheat

The efficacy of *A. salinestris* treatment against VI, under in situ, was evaluated. Our results showed that the VI of maize, sorghum, and wheat seedlings increased significantly when in a combination of *A. salinestris* and *Fusarium* (T3), compared to the control (T1) and when only *Fusarium* (T2) was used in each treatment (Table 2). However, in the treatment where only *A. salinestris* (T4) was used, a three-fold increase in root and shoot length was recorded. 

In our untreated sample (T1), the VI range of maize, sorghum, and wheat, recorded, was 1090.54–1183.67, 1141.18–1572.33, and 2080.62–2144.55, respectively. When only *Fusarium* (T2) was used, the VI range recorded was 161.22–488.13 in maize, 455.06–620.29 in sorghum, and 358.30–670.31 in wheat. The combination of *A. salinestris* and *Fusarium* (T3) exhibited the highest VI of 1829.79 in maize treated with *F. acuminatum.* The lowest VI of 1242.87 was observed in maize treated with *F. culmorum* alone. 

In sorghum, the highest VI of 1449.22 was recorded in the treatment with *F. graminearum* and the lowest VI recorded was 1115.71 in the treatment with *F. equiseti*. In wheat, the highest VI of 2790.64 was achieved in the treatment with *F. nivale* whereas the lowest VI of 1510.53 was recorded in the treatment with *F. culmorum*. When *A. salinestris* was used alone (T4) in maize, sorghum, and wheat, the VI ranges recorded were 2822.76–3550.03, 2841.04–3324.71, and 4106–5423.75, respectively (Figure 3). The VI response in all these treatments was highest in sorghum treated with *A. salinestris,* followed by wheat and maize. 

#### 3.1.4. Effect of *A. salinestris* against Weight of Cereals under Different Treatments

The effect of *A. salinestris* on the total mass of maize, sorghum, and wheat seedlings was also evaluated. In our control (T1), total mass of maize, sorghum, and wheat seedling ranged at 4.48–4.60 g, 2.20–2.29 g, and 2.13–2.23 g, respectively. When treated with only *Fusarium* spores (T2), the total seedlings mass ranged at 2.20–3.35 g in maize, 0.975–2.01 g in sorghum, and 0.95–1.35 g in wheat. In contrast, when treated with both *A. salinestris* and *Fusarium* (T3), a significant increase in the seedlings mass was observed at the extent of 7.38–9.46 g in maize, 2.82–3.19 g in sorghum, and 2.07–3.06 g in wheat, respectively. In the treatment where *A. salinestris* alone (T4) was used, the total seedling mass in maize, sorghum, and wheat seedlings ranged at 11.19–13.96 g, 6.354–7.325 g, and 5.851–6.09 g, respectively (Figure 4). With *A. salinestris* treatment, the total seedlings mass was highest in maize followed by sorghum and wheat seedlings. 

#### 3.1.5. Effect of *A. salinestris* Treatment on Percentage of Germination in Cereal Grains

The effect of *A. salinestris* on germination percentage in maize, sorghum, and wheat was evaluated. In our control (T1), the percentage of germination in maize, sorghum, and wheat was 80.6, 82, and 74%, respectively. In the treatment where *Fusarium* spp. alone (T2) was used, approximately 40% reduction in germination in all the three cereals was observed. In the treatment where a combination of *A. salinestris* and *Fusarium* (T3) was used, a significant increase in germination was observed: 72% in maize, 65% in sorghum, and 66% in wheat. In the treatment where *A. salinestris* alone (T4) was used, the germination percentage reached 98% (Figure 5), and the germination percentage recorded in all the cereals was approx. 91%. 

#### 3.1.6. Effect of *A. salinestris* against Root Colonization of Cereals by *Fusarium* spp.

The extent of resistance provided by *A. salinestris* against trichothecene-producing *Fusarium* spp. was tested by inoculating spores in the rhizosphere of maize, sorghum, and wheat (Table 2), prior to and after transplanting the seedlings in the germination trays as explained above. The total fungal counts of *Fusarium* (10^6^ spores) in treated seeds of maize (T2) were ~6 × 10^7^ CFU/mL, and after *A. salinestris* treatment (T3) to the seedlings, a reduction in fungal count of ~1.2 × 10^7^ CFU/mL was recorded. The range of fungal count obtained with the combination of both *A. salinestris* and *Fusarium* (T3) treatment, among all the cereals, was 2 × 10^7^–4 × 10^7^ CFU/mL. The lowest fungal spore count in the rhizosphere was with *F. sporotrichioides* in maize and wheat and with *F. graminearum* in sorghum. The total spore count of 4.8 × 10^7^–6 × 10^7^ CFU/mL was recorded with *A. salinestris* (T4) treatment in the rhizosphere. Bacteria and fungi were absent in untreated samples (T1) which served as control around the rhizosphere soil of all three cereals tested (Figure 6).

### 3.2. In Planta Studies of A. salinestris against Fusarium *spp*.

Effect of Antagonistic Activity of *A. salinestris* against *Fusarium* spp. 

Experiments were carried out in the greenhouse to assess the antifungal effect of *A. salinestris* against the shoot length of maize, sorghum, and wheat. The shoot lengths in the control (T1) on the 45th day of treatment were 86.30 cm, 86.25 cm, and 64.50 cm in maize, sorghum, and wheat, respectively. The shoot length with *Fusarium* alone (T2) on the 45th day was 53.50–63.26 cm in maize, 44.15–58.55 cm in sorghum, and 34.05–42.05 cm in wheat (Appendix A). There was a significant progressive increase in shoot length in the presence of both *Fusarium* and *A. salinestris* (T3) rather than in the control (T1) and when *Fusarium* spores were used on their own (T2) after 15, 30, and 45 days of germination (Figure 7). The maximum shoot length recorded was also seen in the combination of both *A. salinestris* and *Fusarium* (T3) treatment, with 147.70 cm shoot length in maize infected with *F. culmorum*, 165.90 cm in sorghum infected with *F. graminearum*, and 77.85 cm in wheat infected with *F. sporotrichioides* (Figure 8). 

Root length/total root mass of these cereals was measured at 45 days after growth. The maximum root length/total root mass recorded in our control (T1) was 27 cm/2.41 g, 20 cm/1.61 g, and 16 cm/0.56 g in maize, sorghum, and wheat, respectively. In treatment with *Fusarium* spores alone (T2), root lengths ranged at 22–26 cm in maize, 16–19 cm in sorghum, and 12–16 cm in wheat, respectively. The root mass recorded ranged at 0.69–0.78 g in maize, 1.30–1.60 g in sorghum, and 0.42–0.75 g in wheat, respectively. In our combination of *A. salinestris* and *Fusarium* treatment (T3), we observed a significant root length/total root mass rather than in T1 and T2, and the maximum root length/total root mass recorded was 70 cm/17.53 g in maize, 48 cm/4.52 g in sorghum, and 56 cm/1.90 g in wheat, respectively (Figure 9).

## 4. Discussion

Plant-growth-promoting rhizobacteria (PGPR) can act as an alternative source to chemical fertilizer and it is also eco-friendly and possesses a few beneficial properties. The PGPR bacterial strains are applied to crops of economic value as they stimulate and promote plant growth by biosynthesis of growth regulators at root interface and enhance soil fertility [27,28]. Previous studies indicated PGPR potential in the production of indoleacetic acid (IAA), ammonia (NH_3_), siderophore, and phosphate solubilization [29]. 

Decrease in *Fusarium* count with treatment with PGPR strains was reported previously. PGPR strains of *Azotobacter* spp., *Arthrobacter* spp., *Pseudomonas* spp., and *Bacillus* spp. were screened against fumonisin-producing *F. verticillioides* in maize rhizosphere [30]. In our present study, the antifungal efficacy of *A. salinestris* against trichothecene-producing *Fusarium* spp., isolated from cereals, was observed. Roots are important for the uptake of nutrients for plant metabolism and growth. *A. salinestris*, in our study, enhanced the root activities by increasing length and total root mass. Results of our infection incidence assay (Figure 2) clearly indicated that *A. salinestris* reduced the ability of *Fusarium* spp. to infect and colonize maize, wheat, and sorghum grains. On the other hand, the growth inhibition assay, in dual-culture plates, in which we used ten different species of *Fusarium* (Table 1), indicated that not all *Fusarium* spp. had their growth inhibited by *A. salinestris.* However, in the control (T1) and in the treatment where *A. salinestris* alone (T4) was not inoculated with any *Fusarium* spp., we observed very low infection incidence in all three cereals when compared to the treatment with *Fusarium* spores alone (T2), and in the treatment with a combination of *Fusarium* spores and *A. salinestris* (T3). Our results may be attributed to possible cross-contamination or growth of natural seed-borne microflora, since the grains were not autoclaved for the complete removal of seed-borne microflora. Grains were not autoclaved to maintain the viability of the grains. We also observed a decrease in *Fusarium* concentration in the rhizospheric region of maize, sorghum, and wheat. Chauhan et al. [31] reported the anti-fungal efficacy of *Azotobacter chroococum* against *Rhizoctonia solani* in cotton and rice. Furthermore, *Bacillus* spp., *A. nigrescens* and *Pseudomonas putida* were observed to exhibit antagonistic effect on seed-borne pathogenic fungi and to reduce associated mycotoxin contamination in seeds [17,18]. It was also reported that these bacteria, under greenhouse conditions, enhanced root length, shoot length, percent germination, VI, and mass of the seedlings [17,18]. These reports corroborate our study, in that seed treatment of maize, sorghum, and wheat with *A. salinestris* significantly increased the VI, germination (%), and root and shoot lengths under greenhouse conditions.

In other studies, the ability of *Bacillus amyloliquefaciens* and *Microbacterium oleivorans* in reduction of *F. verticillioides* population and fumonisin production in maize rhizosphere and cobs at field level has been reported [32]. The ability of *Bacillus* spp. in preventing rhizosphere and endorhizosphere colonization of *F. verticillioides* in maize root has also been described [33]. Application of *Bacillus* spp. and *Pseudomonas* spp. has been shown to enhance the growth and yield in maize [34,35]. PGPR *Azotobacter* and other bacterial species, as potent biocontrol agents, positively influence plant-growth-promoting properties in different crops [22,36]. Furthermore, PGPR *Azotobacter* and *Azospirillum* have been shown to promote growth and nutrient uptake in field trials in maize [37]. These reports are in agreement with our present results which show antifungal activity of *A. salinestris* against *Fusarium* spp. in promoting seed germination up to 91% and root and shoot lengths up to 70 cm. In addition, *A. salinestris* was also observed to inhibit *Fusarium* spp. infection on seed treatment in maize, wheat, and sorghum with *A. salinestris*. 

In addition, in our present study, we have demonstrated that pretreatment of maize, sorghum, and wheat seeds with *A. salinestris* can antagonize trichothecene-producing *Fusarium* spp. and inhibit their growth. Our potting experiments, with these cereal seeds, treated with a combination of *Fusarium* spores and *A. salinestris* (T3) under greenhouse conditions, revealed that *A. salinestris* increased total root and shoot lengths compared to *Fusarium* spores when used on its own (T2) and sterile water (T1) treatments. Our findings are in agreement with those of Pereira et al. [32,38]. The authors demonstrated that maize seeds treated with PGPR strains reduced infection incidence by *F. verticillioides* and controlled mycotoxin production at the field level [32,38]. Our results are also in accordance with those obtained by Egamberdiyeva [39], and they also described the synergistic efficacy of PGPR *Pseudomonas alcaligenes*, *Bacillus polymyxa*, and *Mycobacterium phlei* applied to two different soil types with maize, and observed the enhanced stimulatory growth in maize seedlings at different soil conditions [39]. Treatment of maize, wheat, canola, and other cereal crops with PGPR *Azospirillum*, *Bacillus*, *Pseudomonas*, and *Enterobacter* caused a stimulatory effect on crop growth in field and laboratory trials [40,41]. It was also reported that PGPR *Azospirillum brasilense* produces IAA, used as PGPR for enhancing plant growth [42]. Similar results were obtained by Kumar et al. [43] when maize seeds treatment with *Azospirillum* and *Azotobacter* significantly increased plant VI, germination index, and total mass of maize seedlings. Previous reports so far clearly indicate the importance of *A. salinestris* with a prominent attribute of PGPR properties, leading to an increase in root and shoot lengths, VI, and total mass of seedling in cereals. Our present investigation has shown the potential of *A. salinestris* as a biological control agent against toxigenic *Fusarium* spp. Our study, paves the way for an alternate and safe control method in place of synthetic chemicals against *Fusarium* infection and disease management in maize, sorghum, and wheat. Furthermore, this study proved that *A. salinestris* can be successfully used as a seed bioprotectant to control trichothecene-producing *Fusarium* spp. associated with these economically important crops such as maize, sorghum, and wheat which are widely used across the globe. 

## 5. Conclusions

Biocontrol methods are emerging as significant alternatives in place of chemical pesticides for disease management in field crops. Furthermore, application of such biocontrol agents is always safer and more environmentally friendly. In addition, with the use of biocontrol methods, management tools and procedures need to be strictly followed to facilitate the effective preservation of stored commodities, with minimum loss in both quality and quantity [44,45]. Our future study plans to focus on field trials in order to validate our reports on large-scale applications in these economically important crops. 

## Figures and Tables

**Figure 1 jof-08-00473-f001:**
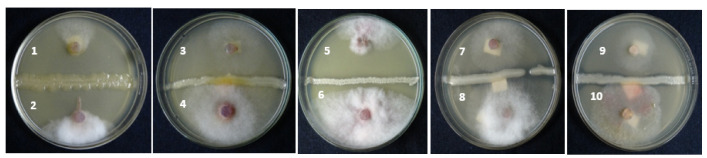
Dual-culture plate assay showing mycelia inhibition by *A. salinestris* against different *Fusarium* spp. Note: (1) *F. sporotrichioides* +++ (2) *F. crookwellense* +++ (3) *F. graminearum* +++ (4) *F. nivale* + (5) *F. culmorum* +++ (6) *F. sambucinum* + (7) *F. poae* ++ (8) *F. avenaceum* + (9) *F. acuminatum* +++ (10) *F.*
*equiseti* ++. [Good inhibition +++; Moderate inhibition ++; Less inhibition +].

**Figure 2 jof-08-00473-f002:**
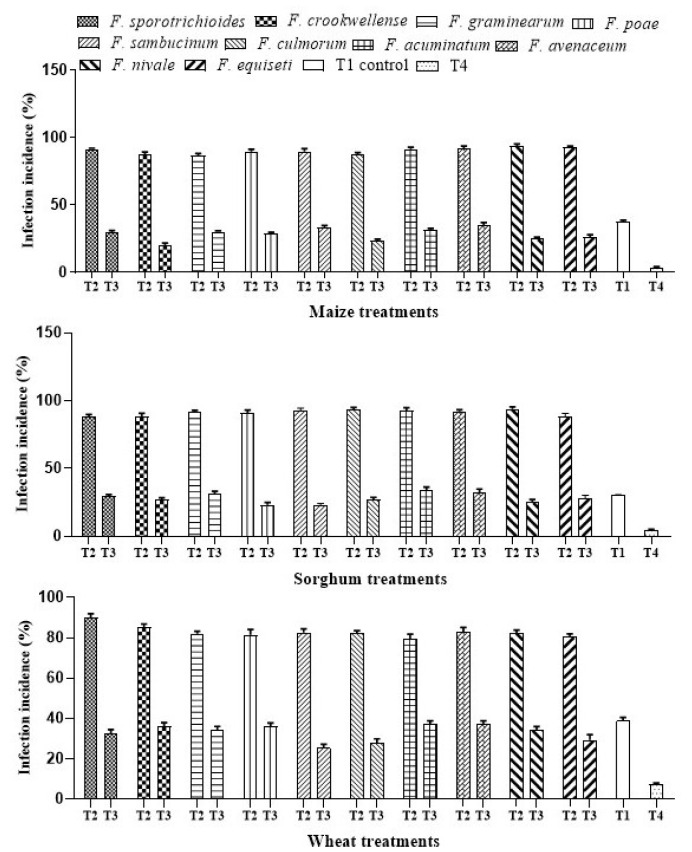
Infection incidence (%) of *Fusarium* in treated seedlings under different treatments. Note: T1—Control, T2—*Fusarium* spp., T3—*A. salinestris* + *Fusarium* spp., T4—*A. salinestris*. Values are mean and deviations of ten samples.

**Figure 3 jof-08-00473-f003:**
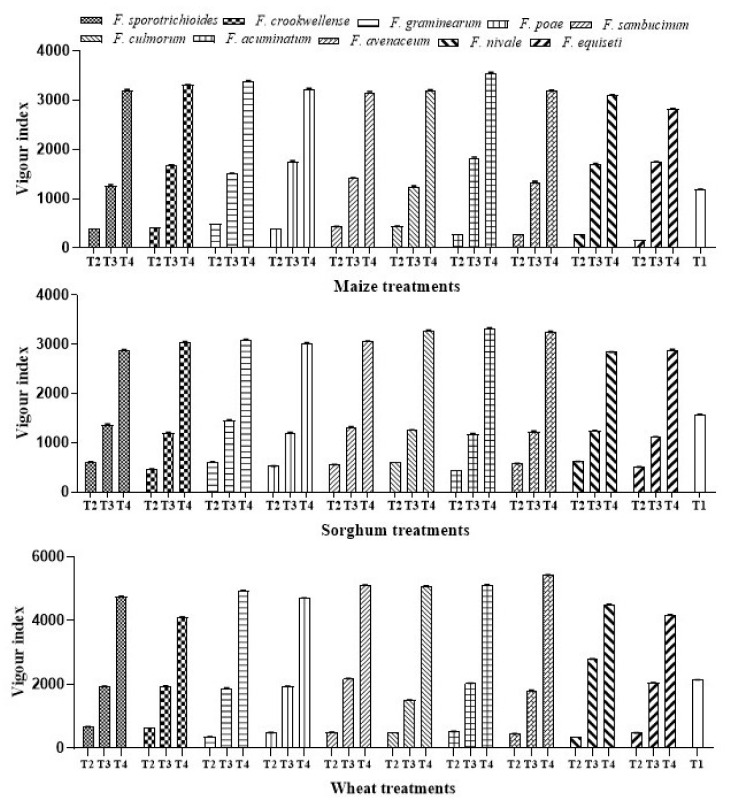
Effect of *A. salinestris* on vigor index (VI) of maize, sorghum, and wheat grains artificially inoculated with different *Fusarium* spp. Note: T1—Control, T2—*Fusarium* spp., T3—*A. salinestris* + *Fusarium* spp., T4—*A. salinestris*. Values are mean and deviations of ten samples.

**Figure 4 jof-08-00473-f004:**
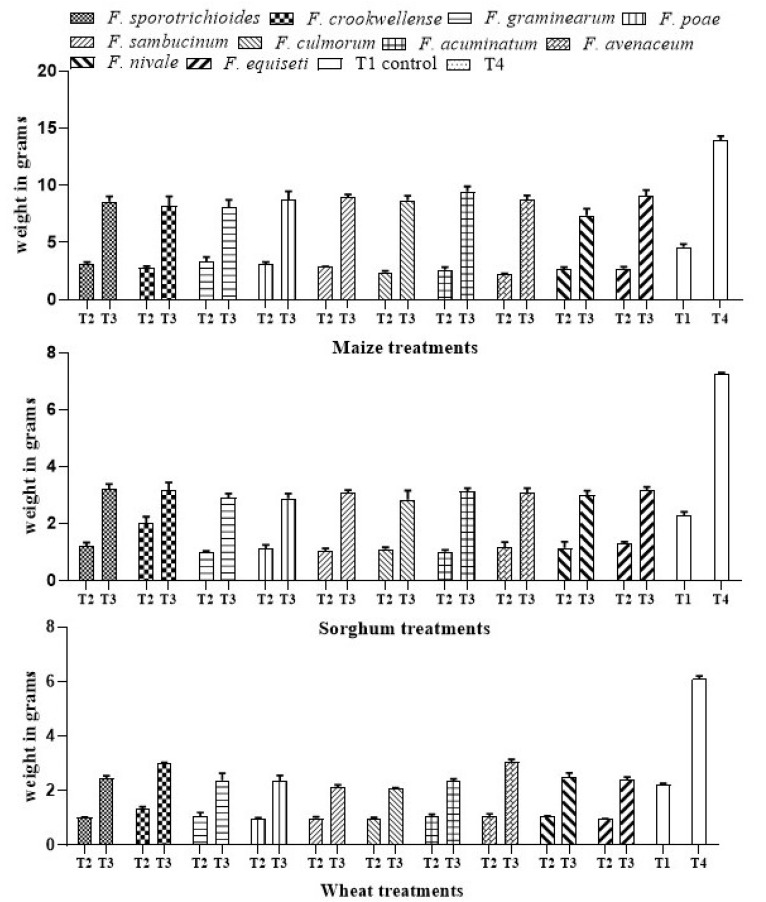
Weight of maize, sorghum, and wheat seedlings under different treatments. Note: T1—Control, T2—*Fusarium* spp., T3—*A. salinestris* + *Fusarium* spp., T4—*A. salinestris*. Values are mean and standard deviations of ten samples.

**Figure 5 jof-08-00473-f005:**
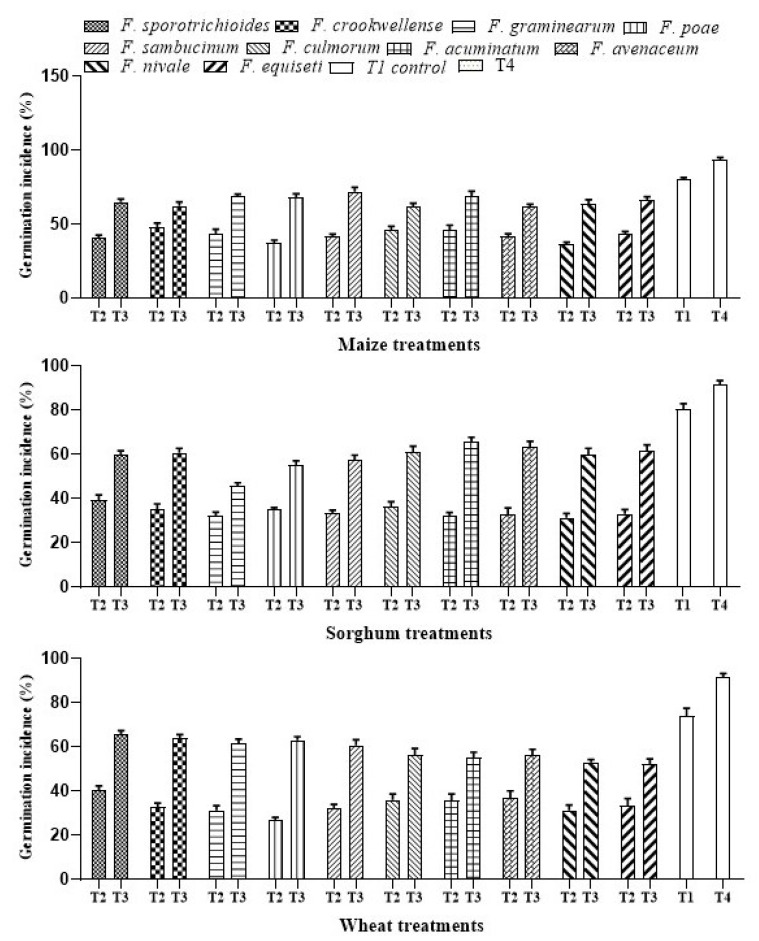
Effect of *A. salinestris* treatment on germination percentage (%) of maize, sorghum, and wheat. Note: T1—Control, T2—*Fusarium* spp., T3—*A. salinestris* + *Fusarium* spp., T4—*A. salinestris*. Values are mean and standard deviations of ten samples.

**Figure 6 jof-08-00473-f006:**
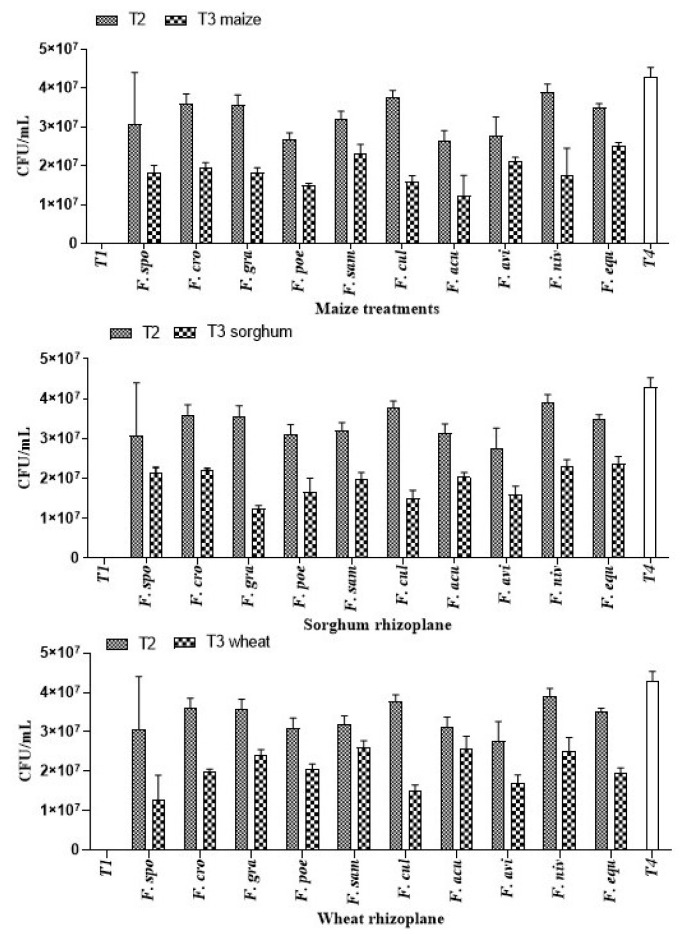
Total count of *Fusarium* spp. isolated from rhizoplane soil of maize, sorghum, and wheat seedlings.

**Figure 7 jof-08-00473-f007:**
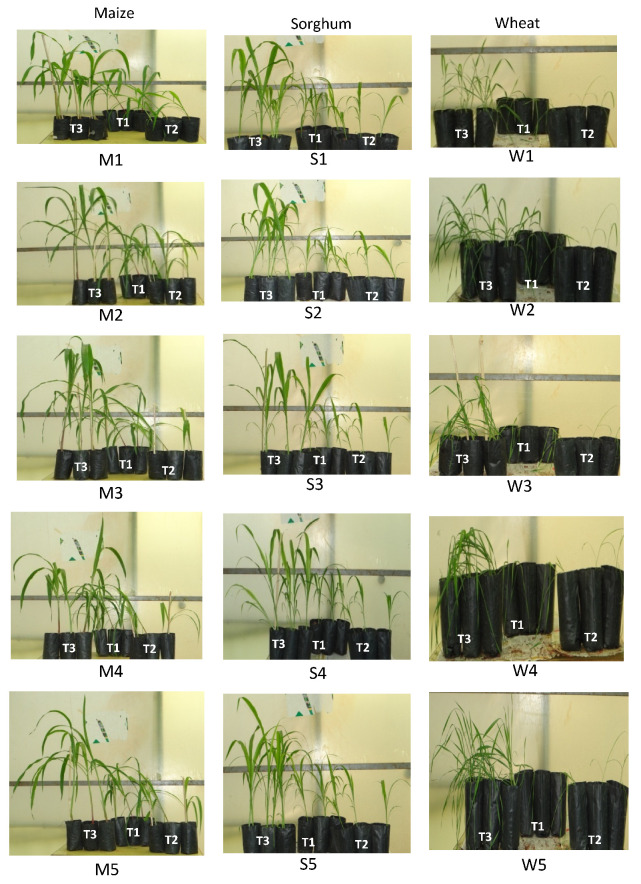
Effect of *A. salinestris* on growth of maize, sorghum, and wheat grains, artificially inoculated with selected *Fusarium* spp. grown for 30 days in potting experiments under greenhouse conditions. Note: M, S, and W represent maize, sorghum, and wheat, respectively. T1 and T2 treatments are the same as described in Table 2, where T3 treatment for M1, S1, and W1 was *A. salinestris* + *F. sporotrichioides*; T3 treatment for M2, S2, and W2 was with *A. salinestris* + *F. crookwellense*; T3 treatment for M3, S3, and W3 was *A. salinestris* + *F. graminearum*; T3 treatment for M4, S4, and W4: *A. salinestris* + *F. poae*; and T3 treatment for M5, S5, and W5 was: *A. salinestris* + *F. culmorum*.

**Figure 8 jof-08-00473-f008:**
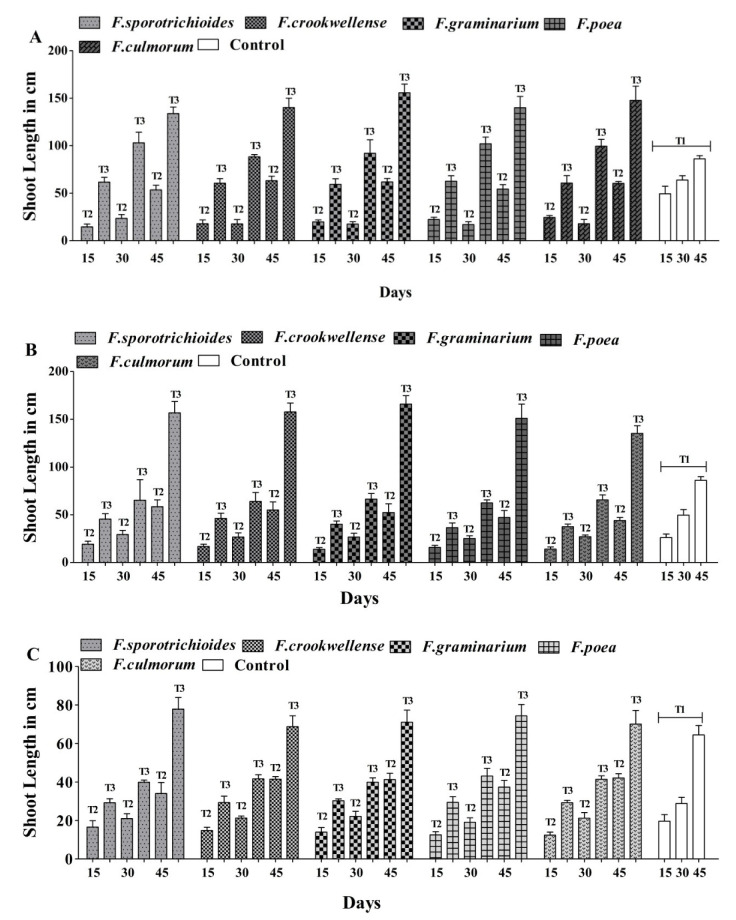
Effect of *A. salinestris* on shoot length of maize, sorghum, and wheat grains under greenhouse conditions after 15, 30, and 45 days of interval. Note: (**A**) maize, (**B**) sorghum, and (**C**) wheat. Values are the means with standard deviation of 10 samples.

**Figure 9 jof-08-00473-f009:**
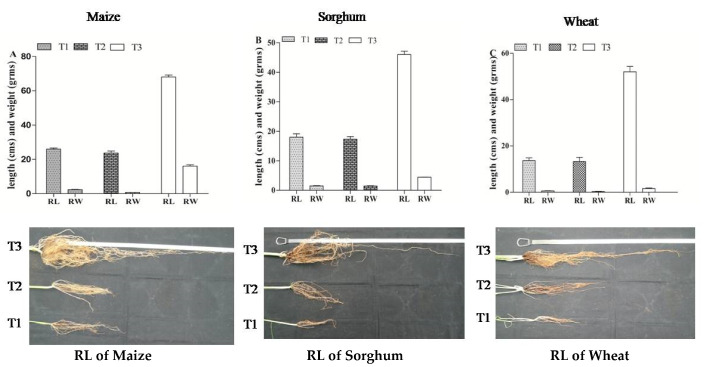
Root length (RL) and root weight (RW) of maize, sorghum, and wheat seedlings after different treatments in greenhouse studies by pot experiments on the 45th day of planting. Note: T1-Sterile water, T2-*Fusarium* spores; T3-*Fusarium* species + *A. salinestris*.

**Table 1 jof-08-00473-t001:** *Fusarium* isolates (Nagaraja et al., 2016 [21]).

Species	Accession Number
*Fusarium sporotrichioides*	KJ371098
*F. crookwellense*	KJ371105
*F. graminearum*	KJ 371099
*F. poae*	KJ 371096
*F. sambucinum*	KJ371095
*F. culmorum*	KJ 371104
*F. acuminatum*	KJ371100
*F. avenaceum*	KJ371102
*F. nivale*	KJ371097
*F. equiseti*	KJ371094

**Table 2 jof-08-00473-t002:** Seed treatments.

Treatment	Treatment Type
T1	Sterile water
T2	Suspension of each *Fusarium* spp. strain at approx. 10^7^ spores mL^−1^.
T3	Suspension of each *Fusarium* spp. strain at approx. 10^7^ spores mL^1^ (T2) + 10^9^ CFU/mL of *A. salinestris*.
T4	*A. salinestris* cell suspension at approx. 10^9^ CFU/mL.

## Data Availability

Not applicable.

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
