# Peer review of "Antifungal Potential of Azotobacter salinestris Strain Azt 31 against Phytopathogenic Fusarium spp. Associated with Cereals"

_jof, 2022, doi:10.3390/jof8050473_

Round 1
Reviewer 1 Report
This paper presents that Azotobacter salinestris Azt 31 is a potential biological control strain against Fusarium species. It can also be used as a seed bioprotectant to reduce fusarium infection to corn, sorghum and wheat seeds. This paper is a topic of interest to the researchers in the related areas but the paper needs a few minor revisions before acceptance for publication, My detailed comments are as follows:
- In Section 2.4.1, the authors mentioned that “The zone of inhibition was measured using a ruler from the edge of the bacterial margin to the edge of each respective fungal colony”. However, according to Figure 1, there are unconspicuous zones of inhibition between bacterial and Fusarium graminearum or bacterial and Fusarium sambucinum, respectively. It would be better if authors can add pictures of Fusarium species control group and explain why graminearum and F. sambucinum were inhibited by Azt 31.
- In Figure 7 and 8, Why only five experiment results of Fusarium species were shown? Please add the reason of selecting these five Fusarium
- In Figure 9 and Supplementary Figure 1, which Fusarium species was used in T2 and T3 treatments.
- There is a significant error in Figure 9. The bar charts of sorghum and wheat do not correspond to the picture. In the picture, the root length of T2 is significantly longer than T1. But in the bar charts, the roots lengths of T1 and T2 are not significantly different.
- Azt 31 can significantly inhibit Fusarium colonization in plant roots, but not all Fusarium are directly antagonized by Azt The inhibitory effect of Azt 31 may also be indirectly antagonized by inducing plant system immunity. I suggest that adding some experiments to detect the plant immunity.
Author Response
Thank you for your appreciation and suggestions to improve our manuscript.

Reviewer 2 Report
I can confirm that the subject matter of this study (Antifungal potential of Azotobacter salinestris strain Azt 31 against phytopathogenic Fusarium spp. associated with cereals) is of interest and relevance for publication in Journal of Fungi. The novelty of manuscript is very high and made focus in a very serious problem. The paper presents original research. Congratulations on a very interesting article. The manuscript is well written, and I have only minor comments to the Authors:
- in keywords change ‘cereals’ on: maize, sorghum and wheat
- Conclusion? - the conclusion should not be a summary of discussion. Make sure the conclusion is short and solid. An idea may be to synthetize in 3-5 bullet the key results of the study, evidences and recommendation. This improvement will increase clearness and readability. Add a practical implications statement.
Author Response
Thank you for your comments on our research.
The mentioned suggestions have been incorporated in the manuscript using track changer.

Reviewer 3 Report
See attach file
You wrote a big paragraph on toxicant even you did not show the results of this in the text
How many replicate you used for each experiment
Are you repeated you experiment this is not clear
Over all the references is too old you can find the new references up to 2022
Most of scientific name are not italic
Some methods need references
Section 2.6 only used ten seeds ?
Fiig 1 needs to revise
You need to added conclusion for your results

Author Response
The paragraph was written as background information, introducing the 10 Fusarium species and toxin production in cereals specially those used in our experiments. In the attached file reviewer’s suggestions has been incorporated throughout the manuscript with track changes.

Round 2
Reviewer 3 Report
in each experiment you need to mention about the number of replicate and if you repeat it or not
in most Fig (2, 3, 4 and 5) the legend in black and white even the columan in color you need to readjust